# Modeling dysbiosis of human NASH in mice: Loss of gut microbiome diversity and overgrowth of *Erysipelotrichales*

**James K. Carter**[1], **Dipankar Bhattacharya**[1], **Joshua N. Borgerding**[2], **M. Isabel Fiel**[3], **Jeremiah J. Faith**[2,4], **Scott L. Friedman**[1]*

**1** Division of Liver Diseases, Department of Medicine, Icahn School of Medicine at Mount Sinai, New York, New York, United States of America, **2** Precision Immunology Institute, Icahn School of Medicine at Mount Sinai, New York, New York, United States of America, **3** Department of Pathology, Icahn School of Medicine at Mount Sinai, New York, New York, United States of America, **4** Department of Genetics and Genomic Sciences, Icahn School of Medicine at Mount Sinai, New York, New York, United States of America

* scott.friedman@mssm.edu

## Abstract

**Data Availability Statement:** All 16S rRNA sequencing files are available in the SRA Bioproject entitled "Modifications to the microbiota in a murine model of NASH" (ID PRJNA668559). All

### Background & aim

Non-alcoholic steatohepatitis (NASH) is a severe form of non-alcoholic fatty liver disease (NAFLD) that is responsible for a growing fraction of cirrhosis and liver cancer cases worldwide. Changes in the gut microbiome have been implicated in NASH pathogenesis, but the lack of suitable murine models has been a barrier to progress. We have therefore characterized the microbiome in a well-validated murine NASH model to establish its value in modeling human disease.

### Methods

The composition of intestinal microbiota was monitored in mice on a 12- or 24-week NASH protocol consisting of high fat, high sugar Western Diet (WD) plus once weekly i.p injection of low-dose $CCl_4$. Additional mice were subjected to WD-only or $CCl_4$-only conditions to assess the independent effect of these variables on the microbiome.

### Results

There was substantial remodeling of the intestinal microbiome in NASH mice, characterized by declines in both species diversity and bacterial abundance. Based on changes to beta diversity, microbiota from NASH mice clustered separately from controls in principal coordinate analyses. A comparison between WD-only and $CCl_4$-only controls with the NASH model identified WD as the primary driver of early changes to the microbiome, resulting in loss of diversity within the 1st week. A NASH signature emerged progressively at weeks 6 and 12, including, most notably, a reproducible bloom of the Firmicute order *Erysipelotrichales*.

additional data files generated for this project can be accessed in the Mendeley dataset Data related to: Modeling dysbiosis of human NASH in mice: Loss of gut microbiome diversity and overgrowth of Erysipelotrichales" (DOI: 10.17632/9kzw8ycw5p.1).

**Funding:** The authors would like to acknowledge the following sources of funding: National Institutes of Health (NIH) T32 GM007280 and T32GM062754 (JKC), NIH DK56621 and the US Department of Defense CA150272 and NIH1P30 CA 196521-01 (SLF), NIH DK124133 and NIH DK112978 (JJF). The funders had no role in study design, data collection and analysis, decision to publish, or preparation of the manuscript.

**Competing interests:** The authors have read the journal's policy and have the following competing interests: JJF is on the scientific advisory board of Vedanta Biosciences. This does not alter our adherence to PLOS ONE policies on sharing data and materials.

## Conclusions

We have established a valuable model to study the role of gut microbes in NASH, enabling us to identify a new NASH gut microbiome signature.

## Introduction

Non-alcoholic fatty liver disease (NAFLD) is a rising cause of chronic liver disease worldwide, affecting nearly one quarter of the global population and rising in parallel with rates of obesity [1, 2]. The disease is comprised primarily of those with steatosis alone, however up to 27% of patients with NAFLD will progress to non-alcoholic steatohepatitis (NASH), in which steatosis is accompanied by inflammation, hepatocyte injury, and fibrosis [3]. NASH can lead to cirrhosis, hepatocellular carcinoma (HCC) or liver failure requiring transplantation [3]. Currently there are no approved therapies for the treatment of NASH, and its pathogenesis has not been clearly defined despite a growing list of potential hepatic and systemic mediators.

The composition of the gut microbiome has been linked to NASH through multiple proposed mechanisms [4–6]. Derangements in the normal microbiota can confer long-term risk of NAFLD, and children born to obese mothers have altered gut microbiota and increased NAFLD risk [7]. Obesity-associated changes to the microbiota promote more efficient dietary energy extraction, accelerating weight gain [8]. Moreover, first-degree relatives of probands with NAFLD-cirrhosis have a 12 times higher risk of advanced fibrosis [9], and although much of the shared risk may be heritable [10, 11], these individuals also typically share some strains of their microbiome.

The intestinal microbiota synergize with genetic, inflammatory, and metabolic drivers of liver injury to promote pathology in NAFLD and NASH. In NAFLD, abnormal microbiota correlate with a leaky intestinal epithelium [12] and compromised gut vascular barrier integrity [13], due in part to altered bile acid signaling [14, 15]. This increases hepatic delivery via the portal circulation of lipopolysaccharides (LPS) and other pathogen-associated molecular patterns (PAMPs), triggering inflammatory innate immune responses and promoting fibrosis through activation of hepatic stellate cells [16]. NAFLD-associated bacteria can also produce TMAVA, a small molecule that impairs lipid metabolism, exacerbating steatosis [17]. *K. pneumonia* identified in the stool of NASH patients has additionally been associated with liver injury through conversion of dietary sugars into ethanol, known colloquially as 'autobrewery syndrome' [18]. Gut microbiota may also accelerate end-stage complications of NASH-cirrhosis by increasing ammonia production to precipitate hepatic encephalopathy [19]. Moreover, microbiome composition may be predictive of NASH severity [20] and progression to cirrhosis [21].

These observations underscore the importance of understanding how the gut microbiome evolves and contributes to NAFLD throughout disease progression. NAFLD-associated microbiota reportedly have reduced species alpha diversity [22, 23]; however, changes to specific taxa have been inconsistent between studies [5]. Well-validated rodent models that faithfully recapitulate human NASH have been limited, but are critical tools to characterize the microbial changes associated with the progression to NASH and to establish any pathogenic role for the microbiota.

We recently developed a murine NASH model that closely replicates the histology and gene expression signature of human NASH in mice by administering a Western Diet (WD) and very low-dose of $CCl_4$ [24]. Compared to others, several features of our model make it superior

for understanding NASH biology, including its reproducible progression to advanced fibrosis in 12 weeks, the presence of associated metabolic abnormalities including insulin resistance, the development of *de novo* HCC, and characteristic histologic features. Therefore, in this study we sought to establish whether our NASH model also captures changes in the gut microbiome associated with human NASH which would enable its use in studies that interrogate its contribution to disease.

## Materials and methods

### Mice

Male C57BL/6J mice 4–6 weeks old were purchased from Jackson Laboratory (Bar Harbor) and maintained on a 12-hour light/dark cycle, housed five per cage in a Helicobacter-free, specific pathogen free (SPF) vivarium. To reduce founder effects, all mice were shipped together and then housed together for a further 72 hours upon arrival at the facility. After this period of co-housing the mice were then randomized to cages. All procedures were approved by the Animal Care and Use Committee of the Icahn School of Medicine at Mount Sinai (IACUC-2018-0060).

### Non-alcoholic steatohepatitis model

To induce NASH, mice were fed *ad libitum* with a high fat, high fructose Western diet containing 21.1% fat, 41% sucrose, and 1.25% cholesterol by weight (Teklad diets, TD. 120528) and a high sugar solution with 23.1 g/L d-fructose (Sigma-Aldrich, F0127) and 18.9 g/L d-glucose (Sigma-Aldrich, G8270). Control mice were maintained on normal chow diet (ND, Lab Diet, Rodent diet 20, #5053) and tap water. A low dose of $CCl_4$ (Sigma-Aldrich, 289116-100ML) 0.2μl/g body weight dissolved in corn oil (10% $CCl_4$/corn oil) or corn oil vehicle control was administered via once weekly intraperitoneal injection. Experimental groups included Control (normal diet, no $CCl_4$ injection), $CCl_4$-only ($CCl_4$ injection and normal diet), Western Diet-only (Western diet, no $CCl_4$ injection) or NASH (Western diet and $CCl_4$ injection). Mice were euthanized at week 12 or 24 using ketamine/xylazine anesthesia followed by exsanguination. Liver and blood samples were collected and processed for downstream analyses of histology, gene expression, and serum parameters. Number of tumors was reported as the number of discrete tumors visible on the liver surface at the time of excision.

### Liver histology

Formalin-fixed, paraffin embedded liver sections were deparaffinized and stained with hematoxylin and eosin (H&E) to assess liver histology or with Picrosirius Red (Sigma, 365548) / Fast Green (Sigma, F7258) for assessment of fibrosis. Stained sections were provided to an expert liver pathologist for evaluation of disease severity utilizing the NAFLD activity score (NAS) and fibrosis stage according to the NASH-CRN scoring system [25]. The pathologist was blinded to the study and treatment arms.

### Fecal microbiota sequencing

Mice were co-housed for one week in a helicobacter-free environment, then randomly divided into treatment groups. Fecal pellets were collected into sterile tubes at weekly intervals, frozen immediately and stored at -80C. DNA was isolated by phenol:chloroform extraction and quantified by Quant-IT Assay Kit (Thermo, Product #Q33120). Microbiota density was calculated as the amount of DNA in a given mass of fecal pellet as previously described [26]. The V4 variable region of the 16S rRNA gene was amplified by PCR and sequenced on the Illumina MiSeq

platform. The 16S rRNA V4 sequences were analyzed in QIIME v 1.9.1 and OTUs were picked against the greengenes reference database 13_8 at 97% identity. Absolute microbial abundance was calculated as the product of relative composition and microbiota density. Diversity indices were analyzed in R with the phyloseq package.

## Serum analysis

Whole blood was collected from the inferior vena cava and allowed to clot at room temperature before centrifugation to separate serum. Serum levels of alanine aminotransferase (ALT) and aspartate aminotransferase (AST) were determined using the VITROS 5,1 FS machine (Ortho Clinical Diagnostics). Endotoxin levels were measured with the Pierce LAL Chromogenic Endotoxin Quantitation Kit (Thermo, 88282). Animals receiving $CCl_4$ were administered the final dose 7 days prior to euthanasia and blood collection.

## Quantitative polymerase chain reaction

RNA was extracted from 30 ug of liver tissue on Qiagen TissueLyser then purified with the RNeasy Kit (Qiagen, 74106). 5 μg of purified RNA was reverse-transcribed into complementary DNA with RNA to cDNA EcoDry (Clontech, 639548). Gene expression levels were determined by qPCR analysis (iQ SYBR Green Supermix, Bio-Rad Laboratories, 1708884) on a LightCycler 480 Real-Time PCR System (Roche). Gene expression was normalized to levels of *Gapdh* mRNA.

Primer sequences used were: *Gapdh* F 5'-CAATGACCCCTTCATTGACC-3', R-5' GAT CTCGCTCCTGGAAGATG-3'; *Col1a1* F 5'-GTCCCTGAAGTCAGCTGCATA-3', R 5'-TGG GACAGTCCAGTTCTTCAT-3'; *Acta2* F 5'-TCCTCCCTGGAGAAGAGCTAC-3', R 5'-TAT GGTGGTTTCGTGGATGC-3'; *Pdgfrb* F 5'-ACTACATCTCCAAAGGCAGCACCT-3', R 5'-TGTAGAACTGGTCGTTCATGGGCA-3'; *16S rRNA* F 5'-TGGCTCAGGACGAACGTGGCGG C-3', R 5'-CCTACTGCTGCCTCCCGTAGGAGT-3'.

## Statistical analysis

Values were compared by two tailed t-test or one-way ANOVA for comparison of two or greater than two conditions, respectively. P values less than 0.05 were considered statistically significant.

## Results

### A reproducible Western Diet and CCl₄-driven murine NASH model

We explored changes to the gut microbiome in our murine NASH model [24]. Mice were fed a high fat, high sugar WD and injected with a low dose of the hepatotoxin $CCl_4$, producing the hallmarks of NASH, including steatosis, inflammation, hepatocyte ballooning, and fibrosis (Fig 1A–1C), all of which contributed to a significantly elevated NAFLD Activity Score (NAS) as determined by a liver pathologist (Fig 1C). Levels of circulating liver enzymes were elevated, consistent with ongoing hepatocyte injury (Fig 1D). The combination of WD and $CCl_4$ produces robust HSC activation marked by increased expression of genes for collagen I, alpha smooth muscle actin (αSMA), and platelet derived growth factor receptor beta (PDGFRβ) (Fig 1E). After 24 weeks on the NASH protocol, all mice spontaneously developed HCC (Fig 1F). NASH mice also display impaired intestinal barrier function, leading to increased leakage of bacterial byproducts such as LPS into the circulation (Fig 1G). This phenomenon has previously been linked to intestinal dysbiosis and hepatic inflammation in NASH [12, 13, 16].

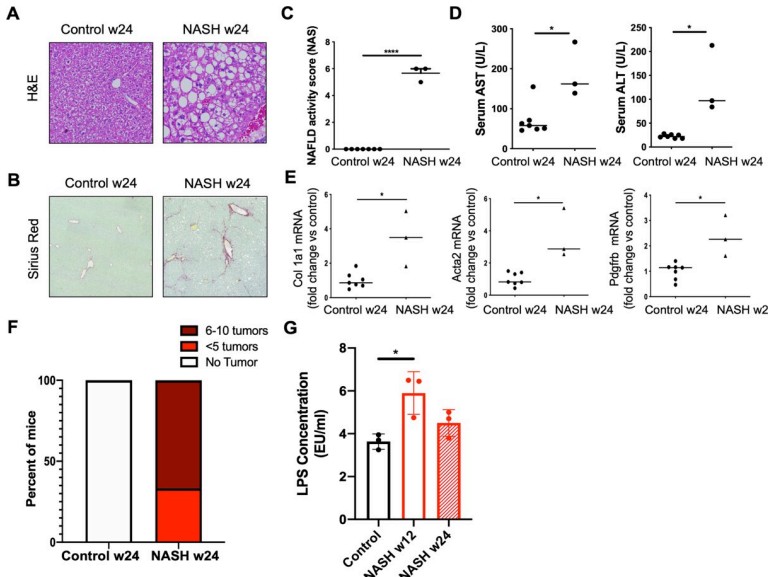

**Fig 1. Hepatocyte injury, fibrosis, and cancer in NASH mice.** Six to eight week old mice were fed regular chow or WD for 24 weeks. NASH mice were injected once weekly with low dose $CCl_4$. Representative liver histology demonstrated by A) H&E stain and B) picosirius red stain for fibrillar collagen to label areas of fibrosis. C) NAFLD activity score (NAS) was evaluated by a blinded pathologist. NASH mice have significantly elevated NAS, indicative of robust disease. D) Liver enzymes were measured in serum collected at the time of sacrifice. E) Expression of selected markers of hepatic stellate cell activation measured by quantitative real-time PCR (qRT-PCR). F) Liver tumors counted on the liver surface at 24 weeks. G) Endotoxin measured in serum collected at 12 or 24 weeks. Image magnification is 200x for H&E (A) and 50x for Sirius Red (B). For (A-F) n = 7 control and n = 3 NASH mice were included. (G) n = 3 mice per group. P values are encoded as * = p < 0.05, ** = p < 0.01, and *** = p < 0.001.

## Murine NASH is associated with depletion and remodeling of the microbiome

To determine whether our model is a useful system to study the gut-liver axis, we characterized the gut microbial community as the NASH pathology progressed. Consistent with previous studies [22, 23], NASH mice had reduced overall microbiota density (0.056±0.019 μg bacterial DNA/ mg feces in NASH vs 0.39±0.2 μg/mg in controls) (Fig 2A), a parameter of the gut microbiome known to influence host physiology [26], and loss of species richness across several measures of alpha diversity. The Faith's Phylogenic Diversity decreased by 18% (p = 0.027) at week 12 and 25% (p = 0.004) by 24 weeks (Fig 2B) with similar losses observed across Chaos Alpha diversity and Observed Species diversity (S1 Fig). We compared the similarity between microbial communities (Beta Diversity) during disease progression using Uni-Frac distances. Principal coordinate analysis (PCoA) could completely resolve NASH samples, indicating a striking remodeling of the microbiome (Fig 2C). This NASH-signature was present at week 12 and remained stable in samples collected at 24 weeks. Together, these results reveal a substantial reorganization of the gut microbiome in this NASH model.

## Major shifts in orders *Erysipelotrichales* and *Bacteroidales* drive the NASH metagenomic signature

In order to better understand how NASH impacts intestinal microbiota, we examined the abundance of specific taxa. Despite an overall loss of bacterial density and reduced alpha diversity, the relative abundance of many bacteria was unchanged by NASH. Instead, large shifts in subsets of bacteria defined the NASH-associated changes (Fig 3A). Most notably, the order

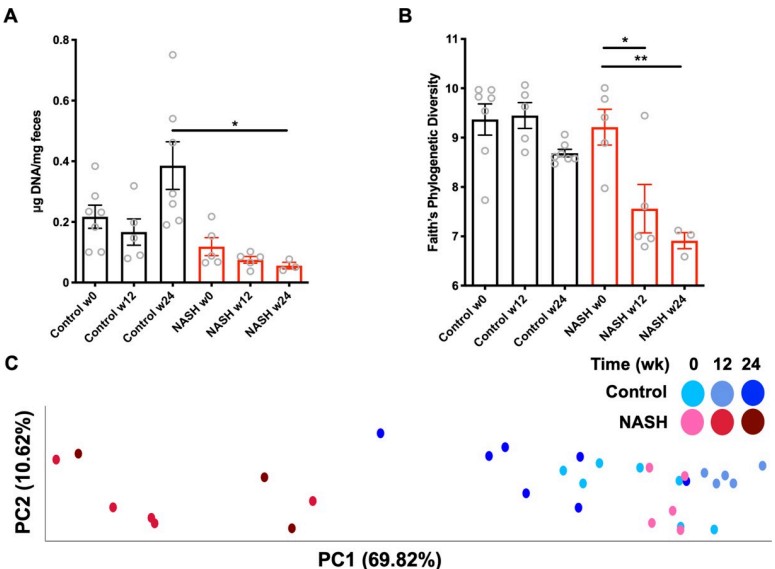

**Fig 2. Loss of abundance and altered composition of microbiome in NASH mice.** Feces were collected from chow-fed controls or NASH mice at weeks 0, 12, and 24. Composition of the microbiome was assessed 16S rRNA amplicon sequencing. A) Microbiota density in feces. NASH mice had significantly lower bacterial density than controls. B) Faith's phylogenetic diversity, calculated to assess species diversity. NASH mice displayed reduced alpha diversity. C) Principal Coordinate Analysis plotting weighted UniFrac distances. NASH mice at 12 and 24 weeks cluster separately from week 0 baseline or chow-fed controls. (A-C) fecal samples from 3–7 mice per condition, per timepoint. P values are encoded as * = p < 0.05, ** = p < 0.01, and *** = p < 0.001.

*Bacteroidales* had significantly reduced abundance compared to chow-fed controls (Fig 3B) and *Erysipelotrichales* underwent a bloom (Fig 3C). The microbiome in control mice was stable across the 24-week study.

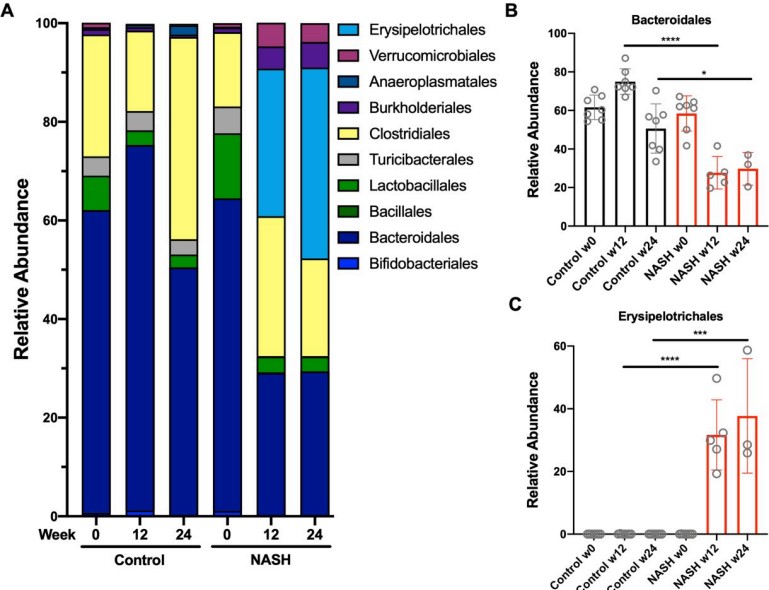

**Fig 3. NASH shifts abundance of *Bacteroidales* and *Erysipelotrichales*.** A) Relative abundance of specific taxa at baseline or after 12, and 24 weeks of NASH protocol. B) *Bacteroidales* abundance is reduced in NASH while C) *Erysipelotrichales* increases dramatically. Fecal samples were sequenced from 3–7 mice per condition, per timepoint. P values are encoded as * = p < 0.05, ** = p < 0.01, and *** = p < 0.001.

## The NASH microbiome signature includes changes independent of Western diet or CCl₄-alone

Diet has a strong influence on gut microbiome composition [27]. We therefore asked whether changes to the microbiome in NASH were driven by feeding the mice a high fat, high sugar WD, by $CCl_4$, or if both interventions were required to elicit the observed changes to the microbiota. To clarify the influences of diet composition and intraperitoneal $CCl_4$ injection on the microbiome, we compared the full NASH protocol to WD-only and $CCl_4$-only mice. The study was conducted for 12 weeks because the microbiome was stable from 12–24 weeks in earlier experiments (Fig 2C). WD consumption had a striking, dominant effect in the first week, causing rapid loss of microbiota density and alpha diversity (Fig 4A and 4B). Structural changes to the microbial population visualized by PCoA of Bray-Curtis dissimilarity (Fig 4C) and weighted UniFrac distances (S6 Fig) captured complete separation of the WD-alone and NASH microbiomes at 1 week. Interestingly, the WD-only group did not display further change after week 1, whereas the microbiome in NASH mice evolved progressively at 6 and 12 weeks, mirroring worsening liver disease (Fig 4C). The microbiota of mice treated with $CCl_4$ injection alone did not change.

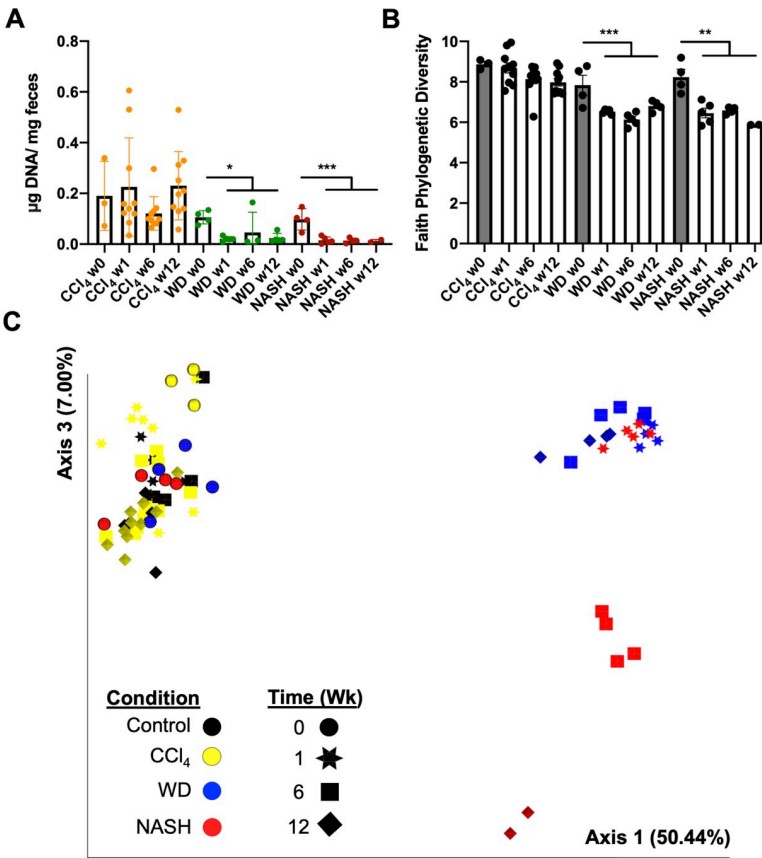

**Fig 4. Separate effects of NASH and WD on the microbiome.** Mice were randomized to control, $CCl_4$-only, WD-only, or complete NASH protocol and followed for 12 weeks. A) Microbiota density decreased in both WD-only and NASH mice, as did B) Faith's Phylogenetic Diversity. C) Principal Coordinate Analysis of Bray-Curtis dissimilarity distances shows changes to beta diversity. WD-only and NASH have separate impacts on population structure. Fecal samples were sequenced from 3–10 mice per condition (mean = 5), except NASH week 12 where n = 2. P values are encoded as * = p < 0.05, ** = p < 0.01, and *** = p < 0.001.

### Expansion of *Erysipelotrichales* is NASH-specific

Exploration of the relative composition of the microbiomes revealed that the substantial expansion of *Erysipelotrichales* is only observed in NASH mice, not those fed WD-alone (Fig 5A and 5B). While *Erysipelotrichales* makes up a tiny fraction of the bacterial population in healthy mice, it grows progressively in NASH mice, making up 12 ±5% of bacteria at 6 weeks and 42 ±12% by week 12 (Fig 5B) versus 2% or less in all other conditions. At week 12 *Erysipelotrichales* overtook *Bacteroidales* to become the most highly represented taxonomic order in the microbiome. Exclusion of *Erysipelotrichales* sequences was sufficient to prevent separation of NASH and control samples in PCoA plots of beta diversity (S7 Fig), underscoring the significant impact this bacterial species has in defining the NASH microbiome. Other changes were observed in both NASH and WD-only cohorts including reduced representation of *Bacteroidales* and *Verrucomicrobiales*, although the loss of *Verrucomicrobiales* between week 1 and 12 in NASH was more extreme (Fig 5C and 5D).

## Discussion

In this study we report a valuable system to interrogate the NASH-microbiome in mice and have characterized NASH-associated alterations to gut microbiota. Remodeling of the microbiome is complete by 12 weeks, coincident with advanced fibrosis, hepatocyte injury, inflammation, and evidence of intestinal barrier dysfunction. The NASH signature persists stably through 24 weeks, during onset of spontaneous HCCs, suggesting a new steady-state for the microbiome in NASH and potentially, a sustained contribution to liver injury. Moreover, the dominant driver of microbiome changes is the high fat diet, whereas $CCl_4$ injections alone did not have a significant effect on the composition of intestinal microbiota.

Consistent with previous reports [22, 23], our NASH model was associated with severe depletion of bacterial species diversity, accompanied by a marked reduction in total microbiota

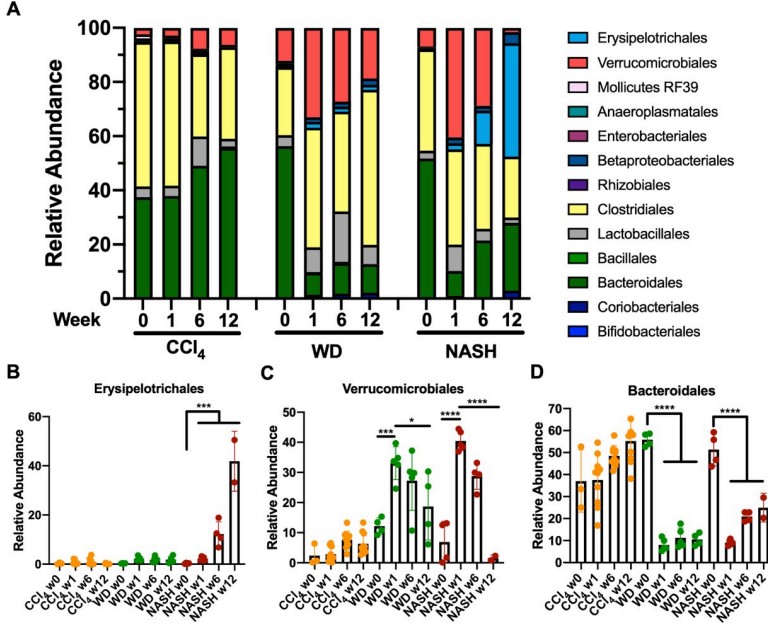

**Fig 5. *Erysipelotrichales* expansion is specific to NASH.** A) Relative abundance of specific taxa over 12 weeks. B) *Erysipelotrichales* bacteria expand in NASH but not other conditions whereas C & D) decreased abundance of *Verrucomicrobiales* and *Bacteroidales* is observed in both WD-only and NASH. Fecal samples sequenced from 3–10 mice per condition (mean = 5), except NASH week 12 where n = 2. P values are encoded as * = p < 0.05, ** = p < 0.01, and *** = p < 0.001.

density. As a result, helpful commensal bacteria may have been lost and pathologic microbes overrepresented. As noted, we can attribute these changes primarily to consumption of high fat, high sugar WD. A decline in alpha diversity and restructuring of the population, indicated by a major shift in beta diversity, take place rapidly within the first week. This corroborates a recent report that proposed increased intestinal permeability, leading to LPS leakage and bacterial translocation, is a critical disease initiating event in a high fat diet model of NASH [13].

We make the novel observation that a bloom of the Firmicute order *Erysipelotrichales* is NASH-specific and not caused by diet alone. *Erysipelotrichales* expansion may be an important feature of NASH microbiota. In a study of the gut microbiome composition in 203 patients, increased abundance of species from the family *Erysipelotrichaceae* was one of the most important bacterial changes predictive of NAFLD-related cirrhosis [23]. Moreover, the abundance of the Firmicute class *Erysipelotrichi* at baseline was positively correlated with the likelihood of developing fatty liver when human volunteers were placed on a choline-deficient diet [28]. Expansion of *Erysipelotrichales* has previously been linked to intestinal inflammation [29]. Furthermore, blocking its growth in a model of parenteral nutrition-associated liver disease attenuated liver injury [30]. In contrast, *Erysipelotrichales* has been identified as a source of the gut-protective short chain fatty acid (SCFA) butyrate in athletes [31], although its behavior could differ in patients with NASH who are more likely to be sedentary and overweight. Previous studies have suggested increases in *Erysipelotrichales* are driven by high fat diet; however, in our model, diet alone is not associated with expansion [32]. This is the first report linking a bloom in *Erysipelotrichales* to NASH in a murine model and future studies should clarify its specific contribution to the disease.

The current study is limited by a number of factors. Although mouse models are the most popular tool for studying the microbiome their intestinal microbiota can be substantially different from humans' [33]. Ours is a descriptive study to validate the NASH model and therefore does not address the causal significance of these changes. The shifts observed could be NASH drivers or passenger effects. Future studies will need to use tools such as FMT, targeted depletion of specific microbes, and gnotobiotic models to establish the impact of changes to gut microbiota on NASH severity.

In summary, we have characterized changes to the gut microbiome in a highly relevant murine NASH model that reinforce its relevance to human disease, and we also describe novel NASH-specific alterations that merit further study. With its disease relevance supported by these findings, this model can be a powerful tool to interrogate the contribution of the gut microbiome to the pathophysiology of NASH.

## Supporting information

**S1 Fig. Multiple measures of alpha diversity are reduced in the intestinal microbiome of NASH mice at 12 and 24 weeks.** Alpha diversity calculated in control versus NASH mice using A) Observed Species or B) Chao Index.
(TIFF)

**S2 Fig. Additional characterization of NASH model at 24 weeks.** A) Images of whole liver collected from control or NASH mice at week 24. B) Serum cholesterol measurements at week 24. C) Mouse weights tracked weekly during the 24 week NASH protocol. D) Liver histology scored by a blinded pathologist.
(TIFF)

**S3 Fig. Comparison of liver histology in NASH versus CCl$_4$ or WD-only at 12 weeks.** A) Liver histology assessed by H&E stain or picosirius red for fibrosis. B) Quantification of Sirius

red positive area to measure extent of fibrosis.
(TIFF)

**S4 Fig. Genus-level shifts in microbiota abundance associated with NASH.** A) Genus-level relative abundance at baseline or after 12, and 24 weeks of NASH protocol. B) Bacteroidales abundance is reduced in NASH while C) Allobaculum strains from the *Erysipelotrichales* order increases dramatically.
(TIFF)

**S5 Fig. Analysis of whole liver tissue bacterial content by qPCR for 16S rRNA.** A) At week 12, RNA was extracted from whole liver tissue and screened for bacterial content by qPCR for the 16S rRNA gene in NASH and control mice.
(TIFF)

**S6 Fig. PCoA of weighted UniFrac distances for control, NASH, Western Diet-only, and CCl₄-only controls.** A) Principal coordinate analysis of weighted UniFrac distances demonstrates changes in bacterial composition in WD and NASH mice.
(TIFF)

**S7 Fig. NASH microbiota do not cluster separately when *Erysipelotrichales* is excluded from the analysis.** Principal coordinate analyses of beta diversity measures A) Weighted UniFrac and B) Bray-Curtis distances show microbiota from NASH and control mice no longer form distinct clusters when *Erysipelotrichales* removed from consideration.
(TIFF)

**S1 Video. Principal coordinate analysis of beta diversity demonstrates separate effects of Western Diet and NASH on microbiome.** Video of principal coordinate analysis of Beta diversity presented in Fig 4C. White = Control, Yellow = CCl₄-Only, Green = WD-Only, Red = NASH; Circle = Week 0, Star = Week 1, Square = Week 6, Diamond = Week 12.
(ZIP)

## Acknowledgments

We thank Gerold Bongers and the Microbiome Translational Center of the Precision Immunology Institute at the Icahn School of Medicine at Mount Sinai for their expertise.

## Author Contributions

**Conceptualization:** James K. Carter, Dipankar Bhattacharya, Joshua N. Borgerding, Jeremiah J. Faith, Scott L. Friedman.

**Funding acquisition:** Jeremiah J. Faith, Scott L. Friedman.

**Investigation:** James K. Carter, Dipankar Bhattacharya, Joshua N. Borgerding, M. Isabel Fiel.

**Writing – original draft:** James K. Carter.

**Writing – review & editing:** James K. Carter, Joshua N. Borgerding, Jeremiah J. Faith, Scott L. Friedman.

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
