## [Decision Letter · Decision Letter 0]

10 Nov 2020

PONE-D-20-32307

Modeling dysbiosis of human NASH in mice Loss of gut microbiome diversity and overgrowth of Erysipelotrichales

PLOS ONE

Dear Dr. Friedman,

Thank you for submitting your manuscript to PLOS ONE. After careful consideration, we feel that it has merit but does not fully meet PLOS ONE’s publication criteria as it currently stands. Therefore, we invite you to submit a revised version of the manuscript that addresses the points raised during the review process. The study has merit.

Please submit your revised manuscript within 60 days. If you will need more time than this to complete your revisions, please reply to this message or contact the journal office at plosone@plos.org. Please include the following items when submitting your revised manuscript:

We look forward to receiving your revised manuscript.

Kind regards,

Gianfranco D. Alpini

Academic Editor

PLOS ONE

Journal Requirements:

'The authors have read the journal's policy and have the following competing interests: JJF is on the scientific advisory board of Vedanta Biosciences'

a. Please confirm that this does not alter your adherence to all PLOS ONE policies on sharing data and materials, by including the following statement: "This does not alter our adherence to  PLOS ONE policies on sharing data and materials.” (as detailed online in our guide for authors http://journals.plos.org/plosone/s/competing-interests).  If there are restrictions on sharing of data and/or materials, please state these.

Please note that we cannot proceed with consideration of your article until this information has been declared.

Reviewers' comments:

Reviewer's Responses to Questions

**Comments to the Author**

1. Is the manuscript technically sound, and do the data support the conclusions?

Reviewer #1: Partly

2. Has the statistical analysis been performed appropriately and rigorously? 

Reviewer #1: Yes

3. Have the authors made all data underlying the findings in their manuscript fully available?

Reviewer #1: Yes

4. Is the manuscript presented in an intelligible fashion and written in standard English?

Reviewer #1: Yes

5. Review Comments to the Author

Reviewer #1: The authors expand on their prior 2018 publication of a murine NASH model to analyze and characterize microbial composition or dysbiosis compared to control mice. The authors have demonstrated that their murine model of WD with CCL4 IP injection can lead to altered microbiome composition and reduction of helpful microbiota. This manuscript will increase the base of knowledge on the incredibly complex influence of the gut microbiome on hepatic damage. Despite great effort and very interesting data, the authors should focus or consider some major and minor points written below:

1. Many of the figures and their legends are not clear on the age of NASH or the number of mice utilized to make these analyses. Please rewrite these and repeat the experiments necessary to increase rigor of the data if the numbers are below 3.

2. The authors make notes of microbial composition to an extent but no analysis of microbial metabolites were performed. To further assess the role of NASH development on the perpetuated microbial composition change metabolomics analysis though HPLC or mass spec should be employed. This may be telling as to why the 12wk NASH microbiota were separated from the 6-12wk WD microbiota visualized on Fig 4C.

3. Although there was great care not to be repetitive of the 2018 manuscript, there has to be further explanation and establishment of the NASH model in this manuscript. There must be demonstrated hepatocyte ballooning, hepatic inflammation, HSC staining, and inflammatory marker expression to convince the audience that this model is indeed representative of NASH.

4. Magnification of images must be stated in the figure legend.

5. Microbial genetic material should be measured in hepatic tissue (as it was in fecal material) to assess the bacterial translocation severity in NASH vs controls. This will help increase the impact of the LPS levels found in serum along with support the lack of intestinal tight junction analysis cited in the discussion.

6. The authors should attempt to make Fig 4C easier to view for any visually impaired or those with printed publications. This is a beautiful figure but without the video supplemented it is difficult to assess the different ages and groups with the black background.

7. More details on the role of Erysipelotrichales should be increased in the discussion. This is an interesting and novel finding that supports previously published human and murine studies. Emphasis on its potential roles should be mentioned despite not being explored in this manuscript.

6. PLOS authors have the option to publish the peer review history of their article (what does this mean?). If published, this will include your full peer review and any attached files.

Reviewer #1: **Yes: **Vik Meadows

---

## [Author Response · Author response to Decision Letter 0]

11 Dec 2020

Academic Editor

We have checked that our manuscript meets the PLOS ONE style requirements. 

'The authors have read the journal's policy and have the following competing interests: JJF is on the scientific advisory board of Vedanta Biosciences' 

a. Please confirm that this does not alter your adherence to all PLOS ONE policies on sharing data and materials, by including the following statement: "This does not alter our adherence to PLOS ONE policies on sharing data and materials.”

We will adhere to all PLOS ONE policies on sharing data and materials. The indicated statement has been added to our competing interests section. 

We have included the updated Competing Interests statement in our cover letter. 

The titles have been copied directly from the manuscript to the online submission form to ensure they are identical. 

Reviewer #1: The authors expand on their prior 2018 publication of a murine NASH model to analyze and characterize microbial composition or dysbiosis compared to control mice. The authors have demonstrated that their murine model of WD with CCL4 IP injection can lead to altered microbiome composition and reduction of helpful microbiota. This manuscript will increase the base of knowledge on the incredibly complex influence of the gut microbiome on hepatic damage. Despite great effort and very interesting data, the authors should focus or consider some major and minor points written below:

1. Many of the figures and their legends are not clear on the age of NASH or the number of mice utilized to make these analyses. Please rewrite these and repeat the experiments necessary to increase rigor of the data if the numbers are below 3.

We have now added information about the NASH model timepoints (duration of NASH) and numbers of mice used in each experiment. In all experiments the number of mice included in each condition was 5-10. Sequencing was performed on fecal samples from 3 to 7 mice per condition and timepoint in all experiments except NASH w12 in figures 4 and 5, where samples were collected from 5 NASH mice but only 2 samples yielded sufficient DNA for sequencing due to loss of bacterial biomass in the NASH condition. The sequencing results from these two samples are representative of the results obtained from n=3 w12 NASH mice in the separate experiment presented in figures 2 and 3. Therefore our results are robust and reproducible across at least n=5 mice. 

2. The authors make notes of microbial composition to an extent but no analysis of microbial metabolites were performed. To further assess the role of NASH development on the perpetuated microbial composition change metabolomics analysis though HPLC or mass spec should be employed. This may be telling as to why the 12wk NASH microbiota were separated from the 6-12wk WD microbiota visualized on Fig 4C.

To further explore what causes the distinct composition of microbiota in NASH shown in figure 4C, we have performed new analyses which are included as supplemental figure S7. In this figure, we remove Erysipelotrichales sequence groups from our data and show that in the absence of Erysipelotrichales the clustering falls apart. This provides additional evidence that expansion of Erysipelotrichales is a key distinguishing feature of NASH microbiota in our model.

We share the reviewer’s interest in exploring how microbial metabolites change in this NASH model; however, in this case, the addition of metabolomics would be insufficient to form strong hypotheses about the changing microbial composition or potential impacts on liver disease. Currently metabolomics analyses are limited to the sets of metabolites that are already known, ignoring what is likely the larger group of uncharacterized metabolites made by the microbiota. Furthermore, the relationship between these known metabolites and NASH pathogenesis not yet understood, as to date there are no defined associations between specific bacterial metabolites and NASH. As such this type of analysis could be misleading even if associations are found. Finally, even if there was a stronger rationale for performing metabolomics, an additional 24 week study would be required as we do not have sufficient fecal mass from prior experiments to perform such an analysis. 

3. Although there was great care not to be repetitive of the 2018 manuscript, there has to be further explanation and establishment of the NASH model in this manuscript. There must be demonstrated hepatocyte ballooning, hepatic inflammation, HSC staining, and inflammatory marker expression to convince the audience that this model is indeed representative of NASH.

A complete characterization of NASH pathology including specific histologic criteria (hepatocyte ballooning, lobular inflammation, and steatosis) is now provided in a new Supplemental figure S2.

The authors also note that the majority of NASH features requested by the reviewer are presented in figure 1. Specifically, figure 1C provides the NAFLD Activity Score (NAS) rating for NASH mice at week 24 as determined by a blinded pathologist. The NAS is scored based on the severity of lobular inflammation, steatosis, and hepatocyte ballooning (Kleiner, D. E., 2005). Figure 1E provides evidence of hepatic stellate cell (HSC) activation based on increased expression of activated HSC marker genes Acta2, Col1a1, and Pdgfrb measured by qPCR in whole liver tissue.

4. Magnification of images must be stated in the figure legend.

The magnification of images (Figure 1) has been added to the figure legend. 

5. Microbial genetic material should be measured in hepatic tissue (as it was in fecal material) to assess the bacterial translocation severity in NASH vs controls. This will help increase the impact of the LPS levels found in serum along with support the lack of intestinal tight junction analysis cited in the discussion.

An experiment was performed to quantify the presence of bacteria in hepatic tissue based on qPCR for the 16S rRNA gene. RNA was extracted from whole liver tissue and converted to cDNA for qPCR analysis. No bacterial 16S rRNA could be detected in liver tissue from control or NASH mice whereas it could be measured in colon included as a positive control. These data are now included as a supplemental figure S5.

6. The authors should attempt to make Fig 4C easier to view for any visually impaired or those with printed publications. This is a beautiful figure but without the video supplemented it is difficult to assess the different ages and groups with the black background.

Figure 4C has been remade to improve readability in printed format and for those with visual impairments. Specifically, the background has been changed from black to white and the condition color scheme has been modified to eliminate Red-Green contrast which might have been difficult for some readers to distinguish. 

For completeness, we include both PCoA of Bray Curtis dissimilarity distance (Fig 4C) and PCoA of Weighted UniFrac distance (supplementary figure S6). 

7. More details on the role of Erysipelotrichales should be increased in the discussion. This is an interesting and novel finding that supports previously published human and murine studies. Emphasis on its potential roles should be mentioned despite not being explored in this manuscript.

We have expanded the discussion of potential roles for Erysipelotrichales although very little work has been done on this specific group of bacteria. In particular we note evidence that high levels of the Firmicute class Erysipelotrichi were associated with increased risk of developing fatty liver when human subjects were placed on a low choline diet (Spencer M, 2011). Although additional study is needed, this could suggest that increases in this taxa may predispose patients to NASH. We also add the contrasting observation that Erysipelotrichales is a source of beneficial short chain fatty acids in physically fit individuals (Estaki M, 2016).

---

## [Editor Report · Decision Letter 1]

16 Dec 2020

Modeling dysbiosis of human NASH in mice: Loss of gut microbiome diversity and overgrowth of Erysipelotrichales

PONE-D-20-32307R1

Dear Dr. Scott L. Friedman,

We’re pleased to inform you that your manuscript has been judged scientifically suitable for publication and will be formally accepted for publication once it meets all outstanding technical requirements.

Kind regards,

Gianfranco D. Alpini

Academic Editor

PLOS ONE
---

## [Editor Report · Acceptance letter]

23 Dec 2020

PONE-D-20-32307R1 

Modeling dysbiosis of human NASH in mice: 
Loss of gut microbiome diversity and overgrowth of Erysipelotrichales 

Dear Dr. Friedman:

I'm pleased to inform you that your manuscript has been deemed suitable for publication in PLOS ONE. Congratulations! Your manuscript is now with our production department. 

Kind regards, 

on behalf of

Dr. Gianfranco D. Alpini 

Academic Editor

PLOS ONE